

# Nutritional values of wild edible freshwater macrophytes

Muta Harah Zakaria[1,2], Shiamala Devi Ramaiya[3], Nordiah Bidin[1], Nurul Nur Farahin Syed[1] and Japar Sidik Bujang[4]

[1] Department of Aquaculture, Faculty of Agriculture, Universiti Putra Malaysia, UPM Serdang, Selangor Darul Ehsan, Malaysia

[2] Universiti Putra Malaysia (UPM), International Institute of Aquaculture and Aquatic Sciences (I-AQUAS), Port Dickson, Negeri Sembilan, Malaysia

[3] Department of Crop Science, Faculty of Agriculture Sciences and Forestry, Universiti Putra Malaysia Bintulu Campus, Bintulu, Sarawak, Malaysia

[4] Department of Biology, Faculty of Science, Universiti Putra Malaysia, UPM Serdang, Selangor Darul Ehsan, Malaysia

Corresponding author
Muta Harah Zakaria,
muta@upm.edu.my

## ABSTRACT

**Background**. The social acceptability of wild freshwater macrophytes as locally consumed vegetables is widespread. Freshwater macrophytes have several uses; for example, they can be used as food for humans. This study determined the proximate composition and mineral content of three freshwater macrophyte species, *i.e.*, *Eichhornia crassipes*, *Limnocharis flava*, and *Neptunia oleracea*.

**Methods**. Young shoots of *E. crassipes*, *L. flava*, and *N. oleracea* were collected from shallow channels of Puchong (3°00′11.89″N, 101°42′43.12″E), Ladang 10, Universiti Putra Malaysia (2°58′44.41″N, 101°42′44.45″E), and Kampung Alur Selibong, Langgar (06°5′50.9″N, 100°26′49.8″E), Kedah, Peninsular Malaysia. The nutritional values of these macrophytes were analysed by using a standard protocol from the Association of Official Analytical Chemists. Eight replicates of *E. crassipes* and *L. flava* and four replicates of *N. oleracea* were used for the subsequent analyses.

**Results**. In the proximate analysis, *N. oleracea* possessed the highest percentage of crude protein (29.61%) and energy content (4,269.65 cal g$^{-1}$), whereas *L. flava* had the highest percentage of crude fat (5.75%) and ash (18.31%). The proximate composition trend for each species was different; specifically, all of the species possessed more carbohydrates and fewer crude lipids. All of the species demonstrated a similar mineral trend, with high nitrogen and potassium and lower copper contents. Nitrogen and potassium levels ranged from 12,380–40,380 mg kg$^{-1}$ and from 11,212-33,276 mg kg$^{-1}$, respectively, and copper levels ranged from 16–27 mg kg$^{-1}$. The results showed that all three plant species, i.e., *E. crassipes, N. oleracea*, and *L. flava* are plant-based sources of macro- and micronutrient beneficial supplements for human consumption.

## INTRODUCTION

One of the top ten factors contributing to mortality was the low intake of vegetables and fruits (*Ezzati et al., 2002*). Vegetables are sources of vitamins and minerals for the antioxidant activity that is needed in the diet to meet the daily micronutrient requirements

(*Gupta & Bains, 2006*). To reduce individual risk and cardiovascular disease, humans and animals require optimal intakes of minerals such as potassium, sodium, calcium, magnesium, copper, manganese, iodine, and zinc (*Mertz, 1982*). To perform physiological functions, micronutrients such as copper, zinc, and iron obtained from food are required in the human body in limited amounts (typically less than 100 micrograms per day). Micronutrient deficiency (*e.g.*, zinc deficiency) causes decreased taste acuity, slow wound healing, impaired development, decreased sexual maturity, impaired immune system function, and impaired metabolism and homeostasis disorders of the thyroid gland (*Almatsier, 2006*).

Freshwater macrophytes are aquatic plants submerged, emerging, or floating on the water surface (*Lacoul & Freedman, 2006*). They occur in swamps, peatlands, lakes, streams, ponds, rice fields, and drainage canals (*Den Hartog, 1981*; *Muta Harah et al., 2005*). Wild plants have played a crucial role in the human diet, and some communities still depend on these wild foods (*Tbatou et al., 2018*). According to *Grubben, Siemonsma & Kasem (1994)*, of the 225 vegetables in Southeast Asia, approximately 100 species are wild weeds. In East Malaysia (*i.e.,* Sarawak), some 43–48 species of wild freshwater macrophytes belonging to 28 families that are considered as weeds are collected and used as edible food and food preparation, medicine, household items for pillows and mats, and even used to make souvenirs (*Muta Harah et al., 2005*; *Muta Harah, Japar Sidik & Suzalina Akma, 2014*). The local collectors also offer freshwater macrophytes for sale in the local markets. Wild freshwater macrophytes are slowly being well received as consumed vegetables (*Muta Harah et al., 2005*; *Saupi, Zakaria & Bujang, 2009*; *Muta Harah, Japar Sidik & Suzalina Akma, 2014*; *Noorasmah et al., 2015*; *Noorasmah et al., 2016*). In addition to being cheaper, vegetarian products are also important as sources of essential minerals in human nutrition (*Saupi, Zakaria & Bujang, 2009*; *Noorasmah et al., 2015*; *Caunii et al., 2010*). Some indigenous leafy vegetables, nuts, and wild fruits also provide energy and are food supplements with good levels of carbohydrates and other nutrients (*Achinewhu, Ogbonna & Hart, 1995*).

Global food problems have challenged all organizations and researchers to investigate the possibility of using wild plant species as supplementary sources of nutrients (*Abubakar et al., 2021*). Wild plants can provide minerals, vitamins, proteins, phenolics, carotenoids, and carbohydrates (*Seal, Pillai & Chaudhuri, 2017*; *Ghanimi et al., 2022*). Studies on the nutritional potential of some wild edible plants have demonstrated their comparability or even superiority to domesticated crops (*Shad, Shah & Bakht, 2013*). Global dietary guidelines recommend increased consumption of fruits and vegetables to mitigate the threat of diet-related diseases, including metabolic disorders, cancer, and cardiovascular diseases (*Stratton et al., 2021*). Therefore, the promotion of these plants will ensure important nutritional sources for food security and sustainable development.

Macrophytes have been used locally in Malaysia as food sources (*Saupi, Zakaria & Bujang, 2009*; *Muta Harah, Japar Sidik & Suzalina Akma, 2014*; *Noorasmah et al., 2015*; *Noorasmah et al., 2016*). The common macrophyte species that are highly ingested by local residents include *Eichhornia crassipes* (Mart.) Solms, *Limnocharis flava* (L.) Buchenau, and *Neptunia oleracea* Lour. (Fig. 1). *Eichhornia crassipes*, which is known as water hyacinth,

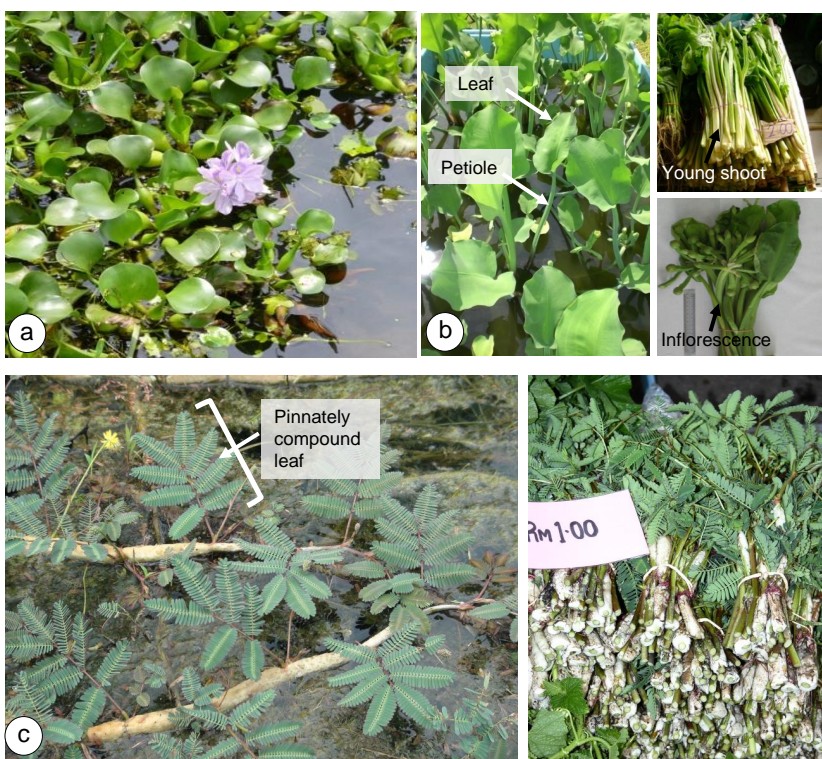

**Figure 1** **Freshwater macrophytes in study sites.** (A) *E. crassipes* from a shallow channel of Puchong, Selangor, Malaysia, (B) *L. flava* from a shallow channel of Ladang 10, Universiti Putra Malaysia, Selangor, Malaysia. Young shoots and inflorescences of (A) and (B) are consumed cooked as vegetables, (C) *N. oleracea* from a shallow channel of Kg. Alur Selibong, Langgar, Kedah. The young leaves' shoot tips are usually consumed blanched or cooked as a vegetable.

is a floating freshwater macrophyte with broad leaves above the water surface and spongy stalks (Fig. 1A). *Limnocharis flava* (yellow velvetleaf) is a plant with pale green leaves and stalks with triangular petiole leaves (Fig. 1B). The leaf blade is papery and broadly ovate-elliptic. Moreover, water mimosa, which is known as *N. oleracea*, has sensitive leaves when touched. It has white spongy air tissues on stems floating on the surface of freshwater (Fig. 1C). Under uncontrolled conditions, these species cause problems in human-made water bodies. Due to their fast rate of reproduction in vegetative and generative states, these plants are commonly referred to as noxious weeds. Rather than destroying them with herbicides, which may affect the ecology, it is preferable to collect and consume them or to use them as for feed.

Local people have been harvesting the young leaves with petioles and inflorescences of *E. crassipes* and *L. flava*, young leaf shoot tips, and immature pods of *N. oleracea* to be consumed as blanched or cooked vegetables, as well as to be sold in local markets to earn income. This scenario is also supported by studies on ethnobotanical information (*Saupi, Zakaria & Bujang, 2009*; *Muta Harah, Japar Sidik & Suzalina Akma, 2014*; *Saupi et al., 2020*) that investigated the most consumed wild aquatic plant species, particularly in
the Bintulu and Sarawak communities. Based on the data, young shoots of *L. flava* and *N. oleracea* were commonly used in dish preparations due to their good palatability and sweet taste with great nutritional quality; however, these species are the least frequently available in the market. Due to the fact that the chemical compositions of aquatic macrophytes vary greatly depending on their species, seasons, habitat, and geographic location, proximate analyses and mineral studies are crucial in determining their nutritional value for future usage possibilities. In addition, a lack of documentation has been published on the nutritional profiling of these macrophytes. Therefore, the present study aimed to determine the proximate composition and mineral content of the commonly consumed wild freshwater macrophytes of *E. crassipes*, *L. flava*, and *N. oleracea*. The findings of this study may also suggest that these species should be commercially grown as new vegetable crops.

## MATERIALS & METHODS

### Sample collection and preparation

Plants of *E. crassipes*, *L. flava*, and *N. oleracea* were collected from shallow channels of Puchong (3°00′11.89″N, 101° 42′43.12 ″E), Ladang 10, Universiti Putra Malaysia (2°58′44.41 ″N, 101°42′44.45″E), and Kampung Alur Selibong, Langgar (06°5′50.9″N, 100°26′49.8″E), Kedah, Peninsular Malaysia, from January to March 2020. The complete specimen of each species was directly arranged on the drawing block, labelled, and pressed as soon as possible by using wooden presses and ensuring that the straps were tight enough to bind the press together. The herbarium samples were examined to verify the identity of the specific plant that was used in a study. The morphological characterization of the aquatic macrophytes was performed following the guidelines based on *Soerjani, Kostermans & Tjitrosoepomo (1987)*.

Young shoots (light green with the tender shoot) were sampled and stored in a ziplock plastic bag accordingly before being kept in an ice chest for transportation to the laboratory. Any adhering materials were removed from the plant samples by washing them with distilled water. Approximately 500 g of fresh samples were cut into small pieces and oven-dried at 60 °C until they reached a constant weight. Dry samples were ground by using an IKA A11[®] Basic Analytical Mill (Ika, Stauffen, Germany) and passed through a 0.2 mm laboratory sieve. The sample powder was labelled and kept inside of an airtight container at room temperature prior to proximate composition (ash, moisture, crude lipid, crude protein, crude fibre, and energy) analysis and mineral content (nitrogen (N), phosphorus (P), potassium (K), calcium (Ca), magnesium (Mg), copper (Cu), manganese (Mn), zinc (Zn), and iron (Fe) analysis). Eight replicates (petiole and leaves) of *E. crassipes* and *L. flava* and four replicates of the whole pinnately compound leaf of *N. oleracea* were used for the subsequent analyses.

### Proximate composition analysis of the wild edible freshwater macrophytes

Proximate analyses of crude fibre composition, crude lipid, crude protein, ash, and moisture amount for the freshwater macrophytes were identified through the standard approaches

of the *AOAC (2000)*. The young shoot moisture was identified by placing the weighed fresh samples in an oven overnight or until a fixed weight was obtained at 60 °C, and the dried mass was determined. For ash determination, the initial crucible weight, which was already labelled and oven-dried (105 °C for 30 min), was measured. Samples (2 g) were put into the crucible and oxidized inside of a muffle furnace at 600 °C for 6 h. Afterwards, the samples were cooled overnight before being placed into the desiccator for 15 min and weighed until a constant weight was achieved. The ash amount was calculated following method 930.05.

Crude protein content was identified by placing 0.2 g of samples inside of a digestion tube and mixed with one tablet of Kjeltec Cu, 5 g K2SO4 + 0.5 g CuSO4, H2O, and 6 ml of concentrated sulfuric acid. The tube was inserted into the Turbotherm Digestor (Gerhardt, Germany) inside of a fume hood and digested for 2 h. The tube was left to cool for 30 min before being inserted into the Protein Analyser (Foss Tecator 2300 Kjeltec Analyser Unit; Foss Analytics, Hillerød, Denmark). The protein concentration was calculated as the percentage of nitrogen by using a conversion factor of 6.25 following method 955.04.

Petroleum ether from the samples was used to obtain crude lipids. Crude lipid was identified by using the 2055 Soxtec Avanti Manual System, Sweden (method 920.39), whereas crude fibre was calculated *via* acid–base digestion according to method 993.19. The estimation of the present carbohydrate level was performed regarding the difference by subtracting the overall percentage of crude protein, crude lipid, crude fibre, ash, and moisture from the 100% dry weight (DW) basis.

## Mineral content analysis of the wild edible freshwater macrophytes

The mineral contents of five macronutrients, including nitrogen (N), potassium (K), phosphorus (P), calcium (Ca), and magnesium (Mg), and four micronutrients, including zinc (Zn), iron (Fe), copper (Cu), and manganese (Mn), were analysed following the Association of Official Analytical Chemists (*AOAC, 2000*) method. All of the nutrient contents were determined by using AAS (Perkin Elmer 200 Flame Atomic Absorption Spectrophotometer, Waltham, MA, United States). The dried samples were milled to less than one mm in diameter. Additionally, the digestion tube was filled with 0.25 g of sample, and 5 ml of sulfuric acid ($H_2SO_4$) was then added. The tube was rotated until all of the plant material was moistened. The mixture was allowed to stand for at least 2 h. Afterwards, two mL of 30–35% hydrogen peroxide ($H_2O_2$) was added. Subsequently, the tube was placed into a port in a digestion block for 45 min at 285 °C. After 45 min, the tube was removed from the block and cooled for 10 min. $H_2O_2$ (two mL) was added if the sample was cloudy, and this process was repeated until the samples were transparent. The samples were then placed into a volumetric flask, filled with distilled water until a level of 100 mL was reached, and mixed. The sample solution was subsequently transferred into 100 mL plastic vials as a stock solution before analysis *via* AAS.

## Statistical analysis

The results are reported as the mean $\pm$ standard error. The data for proximate composition (Table 1) and mineral content (Figs. 2, 3) were statistically analysed by using SPSS, Statistical

Software Program (IBM corporation, Armonk, NY, USA). Means were compared by using single-factor analysis of variance (ANOVA). Post-hoc Duncan's Multiple Range Test (DMRT, $p < 0.05$) (*Zar, 1999*) was performed if the ANOVA result was significant. In addition, a multiple correlation analysis was performed to determine relationships between the abovementioned variables and freshwater macrophyte species. Principal component analysis (PCA) based on the Bray–Curtis similarity index was statistically analysed by using XLSTAT version 2014 software (*Addinsoft, 2015*, New York, USA) to obtain the relationship between the proximate composition and mineral content of freshwater macrophyte species in this study and for available data for other edible indigenous species.

# RESULTS

## Proximate compositions of edible freshwater macrophytes

The proximate composition of freshwater macrophytes is shown in Table 1. *Limnocharis flava* had significantly higher ash and crude lipid contents of 18.31 ± 0.92% and 5.75 ± 0.84%, respectively. In contrast, *N. oleracea* had a significantly higher crude protein of 29.61 ± 0.11% and an energy value of 4,269.65 ± 31.08 cal g$^{-1}$, whereas the crude fibre content was observed to be high in *E. crassipes* (21.34%). The moisture and carbohydrate contents were comparable between *E. crassipes* and *N. oleracea,* ranging from 10.34–10.82% and 50.76–54.42%, respectively. Generally, carbohydrates had a higher concentration in all of the species, along with lower crude lipids. *Eichhornia crassipes* was categorically represented as carbohydrate >crude fibre >ash >moisture >protein >lipid, and this trend was contradictory to that of *L. flava* (carbohydrate >ash >protein >fibre >moisture >lipid) and *N. oleracea* (carbohydrate >protein >moisture >fibre >ash >lipid).

## The mineral content of edible freshwater macrophytes

Figure 2 shows the macronutrient content in the mineral analysis of the three edible freshwater macrophyte species. All of the macronutrient contents were significantly different ($p < 0.05$) between species(except for the K content). *Neptunia oleracea* had the highest N content (40,380.0 ± 730.6 mg kg$^{-1}$), followed by *L. flava* (23,970.0 ± 3,022.0 mg kg$^{-1}$) and *E. crassipes* (12,380.0 ± 1,129.9 mg kg$^{-1}$). *Eichhornia crassipes* had the highest Ca content (11,863.5 ± 316.4 mg kg$^{-1}$), whereas higher Mg values were demonstrated by *N. oleracea* at 2,616.0 ± 68 mg kg$^{-1}$. The trend of macronutrient content shows that *E. crassipes* and *L. flava* are high in K >N >Ca >Mg >P, and *N. oleracea* is high in N >K >Ca >Mg >P.

The micronutrients were significantly different ($p < 0.05$) between the species. *Eichhornia crassipes* had the highest Cu, Mn, and Zn contents, whereas *N. oleracea* had a higher Fe content, as shown in Fig. 3. The trend of macronutrient content shows that *L. flava* and *N. oleracea* have similar trends with Fe >Mn >Zn >Cu, whereas the *E. crassipes* trend is Mn >Fe >Zn >Cu.

Peer J

**Table 1** Proximate composition of freshwater macrophyte species and compares proximate composition (given as mean) of freshwater macrophyte species and other edible plant species.

| No. | Species | Moisture (%) | Ash (%) | Crude lipid (%) | Crude fibre (%) | Crude protein (%) | Carbohydrate (%) | Energy (cal g$^{-1}$) | Trend |
|---|---|---|---|---|---|---|---|---|---|
| **Freshwater macrophytes** | | | | | | | | | |
| 1 | *Eichhornia crassipes* | 10.34 ± 0.76a (7.90–12.70) | 13.23 ± 0.30b (11.93–14.14) | 1.43 ± 0.26c (0.56–2.39) | 21.34 ± 0.33a (20.16–22.58) | 9.58 ± 0.73c (7.53–11.71) | 54.42 ± 0.42a (52.89–56.05) | 3395.15 ± 26.61b (3253.20–3481.90) | C>F>A>M>P>L |
| 2 | *Limnocharis flava* | 7.99 ± 0.46b (6.60–9.70) | 18.31 ± 0.92a (15.70–20.83) | 5.75 ± 0.84a (2.62–8.07) | 15.33 ± 1.18b (11.48–19.50) | 16.58 ± 2.01b (11.21–22.09) | 44.03 ± 0.91b (40.62–48.29) | 3486.98 ± 151.34b (2996.10–3905.70) | C>A>P>F>M>L |
| 3 | *Neptunia oleracea* | 10.82 ± 0.51a (9.33–11.50) | 7.42 ± 0.04c (7.31–7.51) | 3.48 ± 0.12b (3.13–3.64) | 8.73 ± 0.30c (8.04–9.39) | 29.61 ± 0.11a (29.32–29.87) | 50.76 ± 0.29a (49.95–51.20) | 4269.65 ± 31.08a (4203.80–4353.10) | C>P>M>F>A>L |
| df | | 2 | 2 | 2 | 2 | 2 | 2 | 2 | 2 |
| F-value | | 5.572 | 52.334 | 14.480 | 44.218 | 35.564 | 103.928 | 14.216 | |
| P-value | | 0.014 | 5.4305E−8 | 0.000213 | 1.8342E−7 | 8.4202E−7 | 2.9351E−10 | 0.000235 | |

| No. | Species | Moisture (%) | Ash (%) | Crude lipid (%) | Crude fiber (%) | Crude protein (%) | Carbohydrate (%) | Reference (s) |
|---|---|---|---|---|---|---|---|---|
| 4 | *Limnocharis flava* | 79.34 | 0.79 | 1.22 | 3.81 | 0.28 | 14.56 | *Saupi, Zakaria & Bujang (2009)* |
| 5 | *Neptunia oleracea* | 86.26 | 1.05 | 0.25 | 2.30 | 3.23 | 6.91 | *Noorasmah et al. (2015)* |
| 6 | *Ipomoea aquatica* | 72.83 | 10.83 | 11.00 | 17.67 | 6.30 | 54.20 | *Umar et al. (2007)* |
| 7 | *Ipomoea aquatica* | 51.36 | 2.75 | 0.81 | 1.20 | 1.70 | 42.18 | *Igwenyi et al. (2011)* |
| **Vegetables** | | | | | | | | |
| 8 | *Momordica balsamina* | 71.00 | 18.00 | 2.66 | 29.00 | 11.29 | 39.05 | *Hassan & Umar (2006)* |
| 9 | *Amaranthus hybridus* | 59.30 | 4.10 | 1.20 | 3.40 | 4.60 | 27.40 | *Mepba, Eboh & Banigo (2007)* |
| 10 | *Lycopersicon esculentum* | 56.40 | 3.00 | 1.60 | 2.60 | 3.20 | 33.20 | *Mepba, Eboh & Banigo (2007)* |
| 11 | *Telfaria occidentalis* | 58.70 | 3.10 | 1.40 | 2.60 | 3.20 | 32.00 | *Mepba, Eboh & Banigo (2007)* |
| 12 | *Basella alba* | 54.80 | 3.10 | 0.90 | 2.70 | 4.20 | 34.30 | *Mepba, Eboh & Banigo (2007)* |
| 13 | *Myrianthus arboreus* | 83.90 | 16.40 | 13.10 | 11.60 | 18.75 | 40.15 | *Amata (2010)* |
| 14 | *Moringa oleifera* | 76.53 | 7.13 | 2.23 | 19.25 | 27.51 | 43.88 | *Oduro, Ellis & Owusu (2008)* |
| **Spices** | | | | | | | | |
| 15 | *Zingiber officinalis* | 16.10 | 9.60 | 12.10 | 4.50 | 14.90 | 53.10 | *Achinewhu, Ogbonna & Hart (1995)* |
| 16 | *Piper quineense* | 55.30 | 4.50 | 11.70 | 8.70 | 26.60 | 38.90 | *Achinewhu, Ogbonna & Hart (1995)* |
| **Medicinal plants** | | | | | | | | |
| 17 | *Fagonia cretica* | 9.30 | 16.00 | 10.40 | 25.10 | 7.70 | 31.50 | *Dastagir et al. (2013)* |
| 18 | *Tribulus terrestris* | 8.70 | 15.70 | 9.90 | 15.50 | 13.10 | 37.20 | *Dastagir et al. (2013)* |
| 19 | *Chrozophora tinctoria* | 9.70 | 16.00 | 13.00 | 6.70 | 10.50 | 44.10 | *Dastagir et al. (2013)* |
| 20 | *Ricinus communis* | 9.80 | 16.20 | 12.90 | 23.80 | 16.20 | 21.10 | *Dastagir et al. (2013)* |

Notes.

The varying superscript alphabets in the same column demonstrate the contrasts at $p < 0.05$ (ANOVA, Duncan's Multiple Range Test test). Values are given as mean ± SE and range in parenthesis.

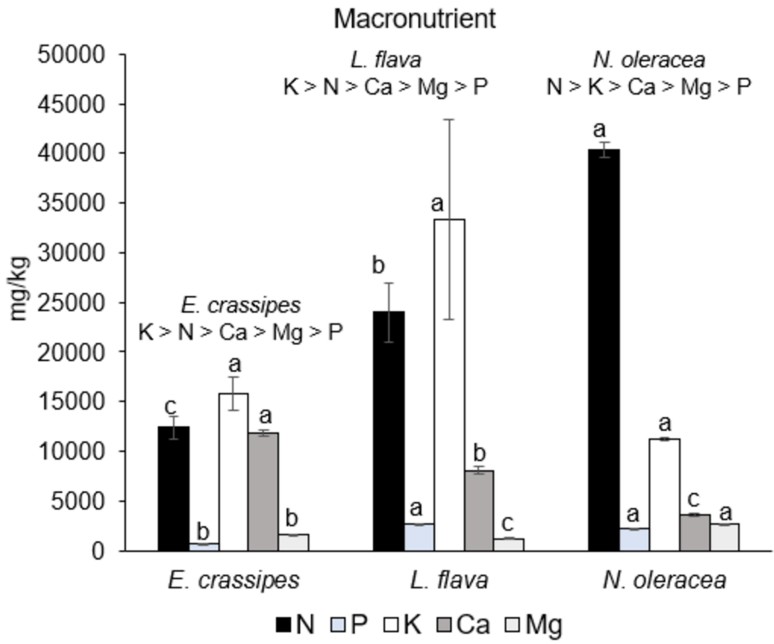

**Figure 2** Macronutrient content in the three wild edible freshwater macrophytes.

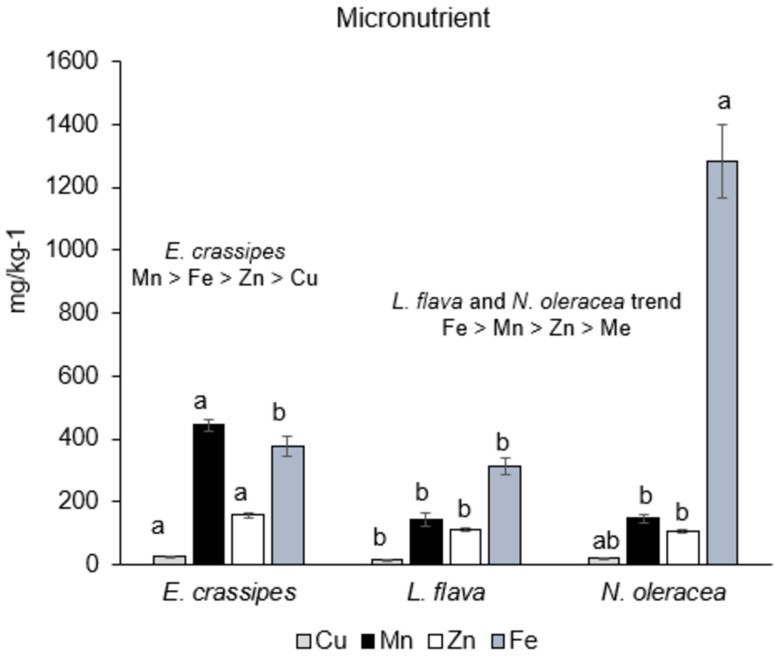

**Figure 3** Micronutrient content in the three wild edible freshwater macrophytes.

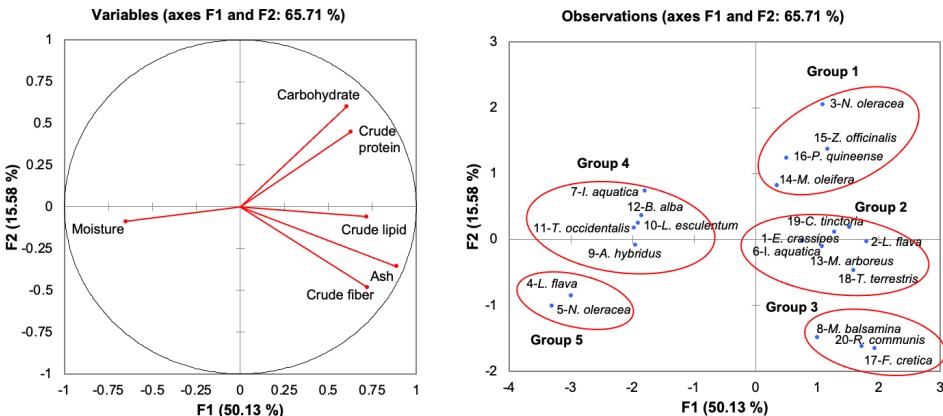

**Figure 4** **Principal component analysis of freshwater macrophytes with other edible plant species based on their proximate composition.** (A) Plot of proximate composition (B) position of PC (principle component) score of species tested with other edible plants according to PC1 and PC2. No. 1–20 represents the assigned number of edible plant species as in Table 1 and corresponds to Groups 1, 2, 3, 4, and 5.

## DISCUSSION

### Proximate composition and comparative analysis with previous studies

Table 1 shows the proximate composition of freshwater macrophytes, and other edible plants listed as vegetables, spices, and medicinal plants that are commonly consumed. The reference species were selected to examine the variation among the freshwater macrophytes in this study and previous studies (no 4–7), as well as among commonly consumed species. The objective of comparing freshwater macrophytes and other commonly consumed species is to demonstrate the potential of aquatic macrophytes as an alternative food to be consumed in daily life. The proximate composition in Table 1 was ordinated with a principal component analysis (PCA). The results in Fig. 4 (based on the Bray–Curtis similarity index at 50% similarity) showed that the total variance of the first two components was 65.71% (PC1 at 50.13% and PC2 at 15.58%). The low value of variance explained by F2 in the PCA is due to the separation of the plants based on their moisture content as defined by their positioning along F1, which represented more than 50% of the total variance of the analyses. The correlation matrix (Table 2) demonstrated a significant moderate correlation between ash and crude lipid ($r = 0.616$) and a strong correlation between ash and crude fibre ($r = 0.767$). In contrast, a moderate negative correlation ($r = -0.525$) was detected between ash and moisture.

There were five distinct groups (1, 2, 3, 4, and 5) displayed in the PCA (Fig. 4). Group 1 consisted of *N. oleracea* in the present study and clustered together with spice plants (*Z. officinali* s and *Piper guineense*) and *Moringa oleifera*, due to higher carbohydrate and protein contents. The current study grouped *E. crassipes* and *L. flava* with *I. aquatica, Myrianthus arboreus, Tribulus terrestris,* and *Chrozophora tinctoria* in Group 2 due to the comparable values of ash and crude lipids. Group 3 had plant species with higher fibre

**Table 2  Correlation matrix for all variables in freshwater macrophyte species with other edible plant species.**

| Variables | Moisture | Ash | Crude lipid | Crude fiber | Crude protein | Carbohydrate |
|---|---|---|---|---|---|---|
| Moisture | 1.000 | −0.525* | −0.378 | −0.269 | −0.249 | −0.366 |
| Ash | | 1.000 | 0.616* | 0.767* | 0.339 | 0.381 |
| Crude lipid | | | 1.000 | 0.322 | 0.399 | 0.273 |
| Crude fibre | | | | 1.000 | 0.347 | 0.237 |
| Crude protein | | | | | 1.000 | 0.431 |
| Carbohydrate | | | | | | 1.000 |

**Notes.**
*Significant at $p < 0.05$.

contents, including *M. balsamina, Fagonia cretica,* and *Ricinus communis.* In contrast, Group 5 consisted of freshwater macrophytes (*L. flava,* and *N. oleracea*), which had higher moisture content. The remaining species were grouped in Group 4 in the positive part of F1 with lower crude protein and carbohydrate contents.

The moisture content ranged from 7.99–10.82% (Table 1) of freshwater macrophytes in this study, which was much lower than the moisture content of 65.02–85.58% and 73.46–77.52% in grasses and sedges, respectively (*Furch & Junk, 1997*). The moisture content of freshwater macrophytes was lower than that of other Nigerian indigenous vegetables, such as *Amaranthus hybridus* (59.30%), *Telfaria occidentalis* (58.70%), and *B. alba* (54.80%) (*Mepba, Eboh & Banigo, 2007*).

The ash content, which is an index of mineral contents in biota, was high in *L. flava* (18.31%) compared to the values reported in young shoots and inflorescence *L. flava* (0.79%) (*Saupi, Zakaria & Bujang, 2009*). However, a similar value was observed for *M. balsamina* leaves (18.00%) (*Hassan & Umar, 2006*). In addition, the crude lipid content was also high in *L. flava* (5.75%), which was higher compared with the same species of *L. flava* at 1.22%, as reported by *Saupi, Zakaria & Bujang (2009)*, as well as for *M. balsamina* leaves at 2.66% (*Hassan & Umar, 2006*). However, the content was lower than in some indigenous wild spices, herbs, fruits, and leafy vegetables, as reported by *Achinewhu, Ogbonna & Hart (1995)*. The crude fibre content of *E. crassipes* was 21.34%, which was slightly lower than that of the green vegetables *M. balsamina* (29.00%) and *Myrianthus arboreus* (11.60%), both of which are consumed as soup in West Africa (*Hassan & Umar, 2006*).

The protein content in *N. oleracea* (29.61%) was comparable to that in *Moringa oleifera* (27.51%) (*Oduro, Ellis & Owusu, 2008*) and *Piper quineense* (26.6%) (*Achinewhu, Ogbonna & Hart, 1995*); additionally, it was higher than that in *M. arboreus* (18.75%) and *M. balsamina* leaves (11.29%) (*Hassan & Umar, 2006*). Higher protein in *N. oleracea* can be related to the excellent source of protein in the Leguminosae species. Plants of the legume family have root nodules in which symbiotic bacteria fix nitrogen to ammonia. This ammonia contains large amounts of proteins and amino acids (*Roos et al., 2020*). The crude protein content of edible *N. oleracea* was 46.37% of the recommended dietary allowance (RDA) (*Institute of Medicine IOM, 2005*). According to *Adeyemi & Osubor (2016)*, *E. crassipes* contains higher nutritional values, especially in their leaf parts, which

consist of concentrated forms of proteins. In addition, water hyacinth leaf protein contains many unsaturated fats, carotenes, xanthophylls, carbohydrates, and minerals, including calcium, iron, and phosphorus (*Kateregga & Sterner, 2007*). Hence, this wild vegetable can be considered a good protein supplement.

Based on the carbohydrate content, higher carbohydrate content was observed in *E. crassipes*, followed by *N. oleracea* and *L. flava*. In a study by *Madsen, Luu & Getsinger (1993)*, water hyacinth roots were actively respiring tissues without any modifications for storing carbohydrates. Therefore, carbohydrates (such as sugar) accumulated in upper plant parts, such as leaf laminae, leaf petioles, and inflorescences, which can explain the higher carbohydrate content in *E. crassipes* compared to others. The carbohydrate content in all freshwater macrophytes (44.03–54.42%) of the current study was higher compared with *L. flava* young leaves and inflorescence (14.56%) that were examined by *Saupi, Zakaria & Bujang (2009)*, as well as *M. balsamina* leaves (39.05%) examined by *Hassan & Umar (2006)* and some Nigerian edible leafy vegetables, such as *Basella alba* (34.30%), *Amaranthus hybridus* (27.40%), and *Lycopersicon esculentum* (33.20%), which were examined by *Mepba, Eboh & Banigo (2007)*. Leafy vegetables are low lipid-containing foods; thus, they are beneficial in several health aspects, such as for avoiding obesity (*Lintas, 1992*). Vegetables usually contain low lipid content in a range of 0.10–0.20%, as reported by *Hazra & Som (2005)*.

Fibre is one of the essential elements when consuming vegetables. Dietary fibre is part of an overall healthy diet to reduce blood cholesterol levels and to decrease heart disease risks and obesity. The fibre content was within the range of herbs (18.71–42.74%), as reported by *Furch & Junk (1997)*. The benefit of consuming vegetables in human nutrition is represented by their high fibre content (*Vadivel & Janardhanan, 2000*). In addition, consuming large quantities of plant vegetables can provide adequate nutrients. *Eichhornia crassipes*, with a high fibre content, is similar to cellulosic wood and other lignocellulosic plants. They are also used as raw materials for papermaking (*Wang et al., 2004*; *Saijonkari-Pahkala, 2001*). A previous study by *Banerjee & Matai (1990)* reported that the leaf part normally had higher fibre content (especially in floating and emergent plants) because they required more strength to support the aerial vegetation. Fibre content is an important dietary component that is widely utilized as a value indicator in poultry and feeding animal diets. For food consumption, high fibre content helps to increase stool volume and decreases the time that waste products spend in the gastrointestinal tract (*Enyi, Uwakwe & Wegwu, 2020*). It has also been reported that a calorific value of more than 12% can be provided by plant food with a good source of crude protein (*Pearson, 1976*). Therefore, *N. oleracea* (29.61%), *L. flava* (16.58%), and other plants, such as *Alisma plantago-aquatica* (14.83%) and *N. nucifera* (14.05%), also meet this requirement.

Based on the National Diets and Nutrition Survey (NDNS) 2014, *Bates et al. (2014)* stated that the average percent carbohydrate uptake is 50% of the food consumed. Carbohydrates (starches) from cereals, roots, and tubers constitute primary energy-giving food, according to *Achinewhu, Ogbonna & Hart (1995)*. Some indigenous leafy vegetables, nuts, and wild fruits constitute energy and can be provided as food supplements because they are also excellent carbohydrates. The energy value in *N. oleracea* (4,269.65 cal g$^{-1}$) was shown to be

high, comparable to that of the local vegetable *Ipomoea aquatica* (3,009.40 cal g$^{-1}$) (*Umar et al., 2007*). In contrast, *Umar et al. (2007)* reported that most vegetables are low in energy value (within the range of 1,250–2,090 cal g$^{-1}$). According to *SACN (2011)*, the energy requirement for 19- to 75-year-old female adults is 1,840–2,175 kcal per day, whereas for 19- to 75-year-old male adults, the requirement is 2,272–2,294 kcal per day.

## Mineral content and comparative analysis with previous studies

Table 3 compares the mineral contents (macro- and micronutrients) of freshwater macrophyte species and other edible plants. The micronutrients (Cu, Mn, and Zn) in Table 4 were ordinated *via* a principal component analysis (PCA), as shown in Fig. 5. Some of the mineral elements were not included for comparison in the PCA, as no data from previous studies were available. The correlation matrix (Table 5) shows a significant moderate correlation between copper and zinc ($r = 0.626$).

Minerals are essential in the diet, even at contents of 4–6% of the human body. The daily body (per day) macro mineral requirement is higher than 100 mg. In addition, they serve as structural components of tissues and for functional cellular and basal metabolism (*Macrae, Robinson & Sadler, 1993*). The contents of N, P, Mg, Ca, Cu, Mn, Fe, and Zn were found to be different among the three plant species studied(except for K). The protein and mineral contents of the water in which the plants are grown are strongly reliant on the composition of the water; for example, the protein and phosphorus contents of the plant are directly proportional to the nutrient loading rate of the water (*Boyd, 1970*). As stated above, higher ash was observed in *L. flava* (18.31%) and *E. crassipes* (13.23%), which contributed to their higher mineral composition. From the current study, nitrogen values ranged from 12,380–40,380 mg kg$^{-1}$. They were higher than those of *M. balsmina* (1,224.9 mg kg$^{-1}$) and *P. quineense* (10,800 mg kg$^{-1}$) but comparable to those of *M. arboreus* (33,800 mg kg$^{-1}$), which showed that freshwater macrophytes are nitrogen-rich vegetables. Moreover, the phosphorus and potassium contents of *L. flava* (2,734.5 mg kg$^{-1}$ and 33,276.0 mg kg$^{-1}$, respectively) were higher than those of *Melochia corchorifolia* (1,018.9 and 72.5 mg kg$^{-1}$, respectively), *M. balsmina* (1,304.6 and 13,200 mg kg$^{-1}$, respectively), and *P. quineense* (300 and 700 mg kg$^{-1}$, respectively). However, its *P* value was lower than that of freshwater *N. oleracea* (4,059.2 mg kg$^{-1}$) and the vegetable *M. arboreus* (5,000 mg kg$^{-1}$) from previous studies by *Saupi, Zakaria & Bujang (2009)* and *Hassan & Umar (2006)*. Additionally, the calcium content in *L. flava* was higher (8,066.5 mg kg$^{-1}$) than that in the same species in prior studies (7,708.7 mg kg$^{-1}$), as well as other freshwater macrophyte species, including *N. oleracea* (3,814.2 mg kg$^{-1}$) and *M. corchorifolia* (7,503.7 mg kg$^{-1}$), and most of the vegetables (except for *M. balsmina*) (9,410 mg kg$^{-1}$). The Mg values of the current study ranged from 1,198.5 to 2,716.0 mg kg$^{-1}$. Furthermore, they were comparable with freshwater macrophytes from previous studies (range between 1,083–2,281 mg kg$^{-1}$) and vegetables of *M. balsmina* and some medicinal plants, such as *A. fragrantissima, A. graveolens,* and *C. bonariensis* (1,200, 1,050, and 1,092 mg kg$^{-1}$, respectively).

Sodium, potassium, phosphorus, and magnesium are macronutrients that play an essential role in calcium homeostasis and bone status (*Heaney, 2015*). Furthermore, fruits and vegetables that are rich in potassium have an alkaline ash characteristic that is important

Peer J

**Table 3  Comparison of mineral contents (macro- and micro-nutrient) of freshwater macrophyte species and other edible plant species.**

| No. | Species | Macronutrients (mg kg⁻¹) | | | | | Micronutrients (mg kg⁻¹) | | | |
|---|---|---|---|---|---|---|---|---|---|---|
| | | N | P | K | Ca | Mg | Cu | Mn | Zn | Fe |
| **Freshwater macrophytes** | | | | | | | | | | |
| 1 | *Eichhornia crassipes* | 12380.0 ± 1129.9c (10040–16720) | 657.0 ± 44.1b (488–824) | 15799.0 ± 1708.9a (11716–27000) | 11863.5 ± 316.4a (10764–13736) | 1649.0 ± 24.7a (1560–1748) | 27.0 ± 2.1a (20–36) | 444.0 ± 18.7a (380–504) | 158.5 ± 8.1a (120–196) | 377.5 ± 31.8b (292-540) |
| 2 | *Limnocharis flava* | 23970.0 ± 3022.0b (12960–35680) | 2734.5 ± 417.1a (1964–5240) | 33276.0 ± 10073.4a (17244–90400) | 8066.5 ± 426.0b (6172–9284) | 1198.5 ± 91.2b (980–1608) | 16.0 ± 0.8b (12–20) | 144.5 ± 20.8b (92–260) | 110.0 ± 4.6b (92–128) | 314.0 ± 25.4b (220-424) |
| 3 | *Neptunia oleracea* | 40380.0 ± 730.6a (38320–41600) | 2275.0 ± 232.7a (1924–2960) | 11212.0 ± 94.3a (11008–11440) | 3583.0 ± 120.2c (3416–3932) | 2616.0 ± 68.0c (2488–2808) | 20.0 ± 2.8ᵃᵇ (16–28) | 146.0 ± 14.3b (124–384) | 107.0 ± 4.7b (100–120) | 1282.0 ± 118.5a (984-1508) |
| | df | 2 | 2 | 2 | 2 | 2 | 2 | 2 | 2 | 2 |
| | F value | 42.76 | 17.71 | 2.77 | 95.10 | 104.26 | 7.33 | 67.56 | 13.99 | 69.04 |
| | *P* value | <0.0001 | 0.0005 | 0.1148 | 0.0001 | 0.0001 | 0.0110 | 0.0001 | 0.0013 | 0.0001 |
| 4 | *Limnocharis flava*[1] | – | – | 42020 | 7708.7 | 2281 | 83.1 | – | 6.6 | – |
| 5 | *Neptunia oleracea*[2] | – | 4059.2 | 32284 | 3814.2 | 1866.7 | 29.7 | 142.3 | 105.3 | – |
| 6 | *Melochia corchorifolia*[3] | – | 1018.9 | 72.5 | 7503.7 | 1083.3 | 335 | 96.8 | 673 | 199.1 |
| **Vegetables** | | | | | | | | | | |
| 7 | *Momordica balsmina*[4] | 1224.9 | 1304.6 | 13200 | 9410 | 2200 | 54.4 | 116 | 31.8 | – |
| 8 | *Amaranthus hybridus*[5] | – | 60 | 450 | 20 | 40 | – | – | 80 | 118 |
| 9 | *Lycopersicon esculentum*[5] | – | 60 | 580 | 850 | 40 | – | – | 50 | 80 |
| 10 | *Telfaria occidentalis*[5] | – | 40 | 280 | 50 | 360 | – | – | 130 | 800 |
| 11 | *Basella alba*[5] | – | 60 | – | 10 | 60 | – | – | 20 | 60 |
| 12 | *Myrianthus arboreus*[6] | 33800 | 5000 | 20130 | 540 | 4600 | 3.550 | 6.957 | 1.82 | 44.125 |
| **Spices** | | | | | | | | | | |
| 13 | *Piper quineense*[7] | 10800 | 300 | 700 | 11500 | 3500 | – | – | – | - |

Zakaria et al. (2023), *PeerJ*, DOI 10.7717/peerj.15496

**Table 3** (*continued*)

| No. | Species | Macronutrients (mg kg⁻¹) | | | | | Micronutrients (mg kg⁻1) | | | |
|-----|---------|---|---|---|---|---|---|---|---|---|
| | | N | P | K | Ca | Mg | Cu | Mn | Zn | Fe |
| **Medicinal plants** | | | | | | | | | | |
| 14 | *Fritillaria us-suriensis*[8] | – | – | – | 355.76 | – | 3.44 | 12.91 | 31.54 | 103.88 |
| 15 | *Gastrodia elata*[8] | – | – | – | 1415.34 | – | 3.42 | 30.09 | 10.64 | 126.15 |
| 16 | *Achillea fra-grantissima*[9] | – | – | – | 14400 | 1200 | 4.44 | 88.5 | 400 | 192 |
| 17 | *Amaranthus virdis*[9] | – | – | – | 15280 | 8255 | 12.44 | 108.1 | 356 | 480 |
| 18 | *Asteriscus graveolens*[9] | – | – | – | 13200 | 1050 | 26.8 | 107.1 | 544 | 204 |
| 19 | *Chenopodium album*[9] | – | – | – | 11500 | 8900 | 14.44 | 148 | 21.2 | 380 |
| 20 | *Conyza bonar-iensis*[9] | – | – | – | 10200 | 1092 | 8.2 | 152.6 | 38.8 | 255.1 |

**Notes.**

The varying superscript alphabets in the same column demonstrate the contrasts at $p < 0.05$ (ANOVA, Duncan's Multiple Range Test test). Values are given as mean $\pm$ SE and range in parenthesis. 1–9 are references list; 1&2- *Saupi, Zakaria & Bujang (2009)*; 3- *Umar et al. (2007)*; 4- *Hassan & Umar (2006)*; 5- *Mepba, Eboh & Banigo (2007)*; 6- *Amata (2010)*; 7- *Achinewhu, Ogbonna & Hart (1995)*; 8- *Yukui et al. (2016)*; 9- *Daur (2015)*.

**Table 4 Comparison of mineral contents of freshwater macrophyte species and other edible plant species.**

| No | Species | Ca | Mg | Cu (mg kg⁻¹) | Mn | Zn | Reference (s) |
|----|---------|-----|-----|------|-----|-----|--------------|
| 1 | *Eichhornia crassipes* | 11,863.5 | 1,649 | 27.0 | 444.0 | 158.5 | Present study |
| 2 | *Limnocharis flava* | 8,066.5 | 1,198.5 | 16 | 144.5 | 110.0 | Present study |
| 3 | *Neptunia oleracea* | 3,583 | 2,616 | 20 | 146 | 107 | Present study |
| 4 | *Neptunia oleracea* * | 3,814.2 | 1,866.7 | 29.7 | 142.3 | 105.3 | *Saupi, Zakaria & Bujang (2009)* |
| 5 | *Melochia corchorifolia* | 7,503.7 | 1,083.3 | 335 | 96.8 | 673 | *Umar et al. (2007)* |
| 6 | *Momordica balsamina* | 9,410 | 2,200 | 54.4 | 116 | 31.8 | *Hassan & Umar (2006)* |
| 7 | *Myrianthus arboreus* | 540 | 4,600 | 3.55 | 6.957 | 1.82 | *Amata (2010)* |
| 8 | *Achillea fragrantissima* | 14,400 | 1,200 | 4.44 | 88.5 | 400 | *Daur (2015)* |
| 9 | *Amaranthus virdis* | 15,280 | 8,255 | 12.44 | 108.1 | 356 | *Daur (2015)* |
| 10 | *Asteriscus graveolens* | 13,200 | 1,050 | 26.8 | 107.1 | 544 | *Daur (2015)* |
| 11 | *Chenopodium album* | 11,500 | 8,900 | 14.44 | 148 | 21.2 | *Daur (2015)* |
| 12 | *Conyza bonariensis* | 10,200 | 1,092 | 8.2 | 152.6 | 38.8 | *Daur (2015)* |

**Notes.**
*Neptunia oleracea* study by *Saupi, Zakaria & Bujang (2009)*.

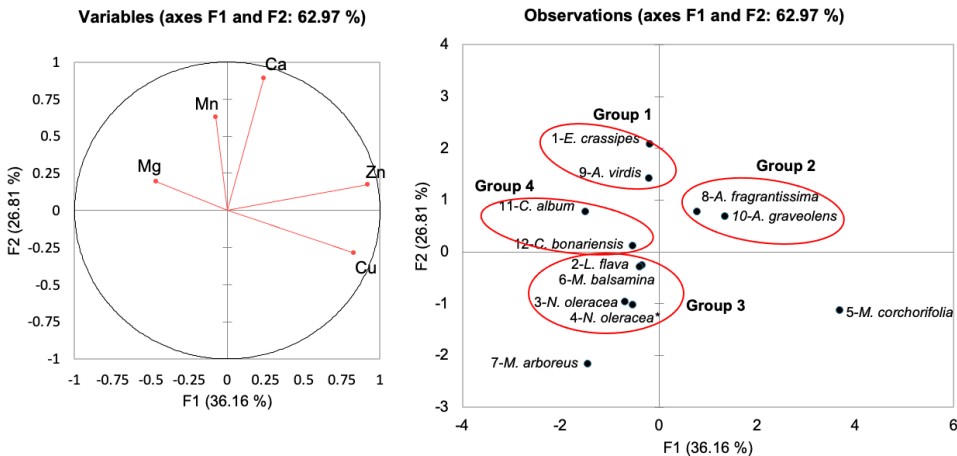

**Figure 5 Principal component analysis of freshwater macrophytes with other edible plant species based on their mineral contents.** (A) plot of mineral contents (B) position of PC score of species tested and other edible plants according to PC1 and PC2. No. 1–12 represents the assigned number of edible plant species as in Table 1 and corresponds to Groups 1, 2, 3, and 4.

in the diet. Adequate phosphorus indicates sufficient protein content in a healthy diet. Both calcium and phosphorus play a role in the growth and maintenance of muscles, bones, and teeth (*Turan et al., 2003*). According to the *National Coordinating Committee on Food and Nutrition (NCCFN) Ministry of Health Malaysia(2017)*, the calcium intake for men and women aged 19–29-years-old was 800 mg, whereas for pregnant women, this intake was was 1,000 mg. Therefore, the value of calcium in this plant indicates that this plant may serve as a vital calcium source.

**Table 5** Correlation matrix for Ca, Mg, Cu, Mn, and Zn in freshwater macrophyte species with other edible plant species.

| Variables | Ca | Mg | Cu | Mn | Zn |
|---|---|---|---|---|---|
| Ca | 1.000 | 0.157 | −0.110 | 0.271 | 0.413 |
| Mg | | 1.000 | −0.246 | −0.164 | −0.242 |
| Cu | | | 1.000 | −0.096 | 0.626[*] |
| Mn | | | | 1.000 | −0.133 |
| Zn | | | | | 1.000 |

Notes.

*significant at $p < 0.05$.

Based on the Bray–Curtis similarity index at 50% similarity, Fig. 5 shows that the total variance of the first two components was 62.97% (PC1 had a total variance of 36.16%, and PC2 had a variance of 26.81%). *Eichornia crassipes* was grouped with a medicinal plant (*A. virdis*) in Group 1 due to higher Mn (444.0 and 108.1 mg kg$^{-1}$, respectively) and similar Cu values (27.0 mg kg$^{-1}$ for *E. crassipes* and 26.8 mg kg$^{-1}$ for *A. graveolens*). Group 2 consisted of medicinal plants, including *A. fragrantissima,* and *A. graveolens,* with higher Zn contents (400 and 544 mg kg$^{-1}$, respectively). The recommended zinc uptake for men and women (19–29-years-old) is 6.70 mg and 4.90 mg, respectively, whereas for pregnant women, this intake is 5.50–10.00 mg (*National Coordinating Committee on Food and Nutrition, NCCFN) Ministry of Health Malaysia(2017)*). Zinc deficiency results in retarded growth and delayed sexual maturation (*Berminas, Charles & Emmanuel, 1998*). Both *L. flava* and *N. oleracea* were clustered in Group 3 with *N. oleracea* and *M. balsmina.* Moreover, *C. album* and *C. bonariensis* were clustered in Group 4, with comparable Mg values ranging from 1,092–8,900 to 196 mg kg$^{-1}$.

Furthermore, iron (Fe) is one of the essential micronutrients in forming haemoglobin and for the functioning of the central nervous system in the body (*Adeyeye & Otokiti, 1999*). The results showed that *L. flava* and *E. crassipes* exhibited iron levels at 314.0 mg kg$^{-1}$ and 377.5 mg kg$^{-1}$, respectively, compared with the iron values in medicinal plants (*C. bonariensis* and *C. album*), which ranged from 255.1–380 mg kg$^{-1}$. The high iron content in the plant could be the reason as to why iron requirements are higher in pregnancy than in the nonpregnant state.

## CONCLUSIONS

Although they are thought to be aquatic weeds due to their fast growth and nuisance to the aquatic environment, freshwater macrophytes are also consumed as food, especially by the local residents, for their subsistence. The results of the current study showed that the species had higher and comparable protein (*L. flava* and *N. oleracea*) and carbohydrate (*E. crassipes* and *N. oleracea*) contents than other edible vegetables that are suitable for human consumption as energy sources. An adequate amount of ash (*L. flava* and *E. crassipes*) and mineral analysis in each plant species can serve as alternatives in food nutrient supplements. These aquatic plants are also useful staple foods because they are simple to cultivate, spread quickly, and require minimal care.

## ACKNOWLEDGEMENTS

We are grateful to AJE for revising and checking the English in this paper.

### Funding

This study was funded by UPM under the Ministry of Education Malaysia (MoHE) with the collaboration of the Japan Society for the Promotion of Science (JSPS), Core-to-core CREPSUM and 6300812 grant funded by Country Garden Pacificview Sdn. Bhd. The funders had no role in study design, data collection and analysis, decision to publish, or preparation of the manuscript.

### Grant Disclosures

The following grant information was disclosed by the authors:
UPM under the Ministry of Education Malaysia (MoHE) with the collaboration of the Japan Society for the Promotion of Science (JSPS).
Core-to-core CREPSUM.
Country Garden Pacificview Sdn. Bhd: 6300812.

### Competing Interests

The authors declare there are no competing interests.

### Author Contributions

- Muta Harah Zakaria conceived and designed the experiments, performed the experiments, analyzed the data, prepared figures and/or tables, authored or reviewed drafts of the article, and approved the final draft.
- Shiamala Devi Ramaiya conceived and designed the experiments, performed the experiments, analyzed the data, prepared figures and/or tables, authored or reviewed drafts of the article, and approved the final draft.
- Nordiah Bidin performed the experiments, analyzed the data, prepared figures and/or tables, and approved the final draft.
- Nurul Nur Farahin Syed analyzed the data, prepared figures and/or tables, and approved the final draft.
- Japar Sidik Bujang conceived and designed the experiments, authored or reviewed drafts of the article, and approved the final draft.

### Data Availability

The raw data is in the Supplementary Files.

### Supplemental Information

Supplemental information for this article can be found online at http://dx.doi.org/10.7717/peerj.15496#supplemental-information.

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
