# Peer review of "Nutritional values of wild edible freshwater macrophytes"

_PeerJ, doi:10.7717/peerj.15496_

## Round 0.1 · original submission · Minor Revisions

Read carefully the reviewers' comments, try to address all the queries and if so can be accepted for publication.

Reviewer 1 ·

Basic reporting

The manuscript describes the chemical composition of three freshwater plant species that are consumed as food by the communities in Malaysia. The authors determined the proximate composition and mineral contents of three macrophyte species namely , Eichhornia crassipes, Limnocharis ûava, and Neptunia oleracea using standard methodologies.

This study represents another routine analysis of vegetables that might have limited applications considering these are widely consumed as food in other parts of the world. The main limitation of this work is the fact that chemical composition of plants change with maturity stages and can be affected by the environment where they grow among others. In this study, samples were collected from different locations but how these factors could have affected the outcome was not given sufficient attention.

Some of the issues that require clarification and explanation:
Line 97-99:. A clear justification for selecting the three species in this study should be provided.

Line 153-163: Description of the plant samples can be summarised in the Introduction section.

There is an issue with data duplication in both different formats (Tables and Figures).
For instance, Table 1 (mean & SE) and Table 2 (only mean) both showed proximate composition.

Data presentation need to be revised.
For instance, Figures 2 & 3: It is unnecessary to provide the raw data alongside the bar charts.

Figure 4 & 5: Please explain the criteria of selection for the additional "wild edible plant species". This could possibly introduced bias in data analysis (refer below).

The discussion section is weak with only citation and description of the numerical values from the literature. There is no attempt to provide possible explanations for the variation in chemical composition between the 3 species used in this study.

Experimental design

Experimental design and procedures used lack clarity in many aspects.

Lines 102-114: The following details are needed.
- When the samples were collected?
- Who authenticated the samples?
- What are the characteristics of "young shoot"?
- What do the authors mean by the "collectors, consumers and sellers' preferences"?
- Why different number of replicates of the 3 species used in this study?

Line 120: Please clarify if fresh samples or air-dried samples were used in moisture determination.

Line 131-137: This section is vague. Please explain the procedures used in mineral content analysis with appropriate citation.

Validity of the findings

The inclusion of additional data from selected "wild edible plant species" is arguable as there is no clear basis for data selection, and the choice of data of which plant samples to be included will influence the outcome of the PCA and correlation analyses as there is variation in the methodology used to collect the data. Findings from this study could have been compared to the literature of the same type of samples (i.e., the freshwater macrophytes). With limited cross reference to previous literature on macrophytes' chemical composition, the work, in my opinion, does not advance our knowledge on their chemical composition.

Lines 365-367: Please draw a more specific conclusion based on the authors' findings.

Additional comments

This study represents another routine analysis of nutrient composition of plant samples. Although some comparisons and data analysis were done with other plant species in the literature, the basis for such comparison is vague, hence, it is unclear what can be deduced from such analyses.

Reviewer 2 ·

Basic reporting

Zakaria et al. have determined the composition and mineral content of three freshwater macrophytes species: Eichhornia crassipes, Limnocharis flava, and Neptunia oleracea. They advocated that N. oleracea possessed the highest percentage of crude protein while L. flava had the highest percentage of crude 37 fat and ash. Overall the paper is good and within the scope of the journal. I have only a few comments;
1. Language of the paper is not up to the mark. Such as example, the legend of figure 1, "Freshwater macrophytes in this study." sentence is incomplete and not making any sense.
2. Authors claim the species under investigation are wild and edible varieties. Do authors document these edible varieties before the study?
3. Material method section needs to elaborate.
4. PC score, please give a full name, at least if it first appears in MS.

Experimental design

Okay

Validity of the findings

Okay

Additional comments

N/A

Reviewer 3 ·

Basic reporting

This paper analyzed the proximate composition and mineral content of three freshwater macrophytes species, Eichhornia crassipes, Limnocharis flava and Neptunia oleracea. With data gathered from this research and from previously published studies, the author also compared proximate and mineral contents using principal component analysis (PCA) with other edible plants. The manuscript is well-written, with clear and professional English, and the background information is organized. For this manuscript, I have a few suggestions for improvement.

Experimental design

1. The authors prepared eight replicates of Eichornia crassipes and eight replicates of Limnocharis flava, four replicates of Neptunia oleracea (Line 112-114) for sample collection and preparation. The manuscript should explain the reason why there were only four replicates for Neptunia oleracea.

2. In addition, from supplementary files, for macro-micro nutrients raw data, the authors showed data in the following samples: Four replicates of Eichornia crassipes leaves, four replicates of Eichornia crassipes petiole; four replicates of Limnocharis flava leaves, four replicates of Limnocharis flava petiole; four replicates of Neptunia oleracea (all).
From line 160-161, the manuscrips showed: Young leaves with petioles and inflorescences of E. crassipes and L. flava are consumed as cooked vegetables. It would be great for the authors to indicate the different parts of the samples that were collected to conduct tests. What’s more, at line 162-163, the manuscript also mentioned: the young leaves shoot tips, and immature pods of Neptunia oleracea are usually consumed blanched or cooked as a vegetable. In order to keep a consistent sample collection, It would be great if the authors can show raw data of immature pods of Neptunia oleracea, or please explain the reason why the tests were done on Neptunia oleracea (all).

Validity of the findings

It would be great if the authors can pay more attention to the result part, some information presented in the result is not consistent with what the figures/tables presented. For example, at lines 178-179, the author mentioned N. oleracea had higher moisture, protein, carbohydrate, and energy contents. The author failed to show a comparison among these three plants, because from table #1, for the carbohydrate level, it showed: Eichhornia crassipes’s carhydrate level is higher than Neptunia aleracea. Another example, at line 192, the author mentioned: Eichhornia crassipes also have the highest Ca and Mg values, however from Fig 2, it showed Neptunia oleracea has the highest Mg value.

---

## Round 0.2 · Minor Revisions

Authors have addressed all of the reviewers' comments.

The manuscript needs English proofreading (mainly for grammar) before publication.

Reviewer 2 ·

Basic reporting

N/A

Experimental design

N/A

Validity of the findings

N/A

Additional comments

N/A

Reviewer 3 ·

Basic reporting

The authors analyzed the nutritional composition of three species of freshwater macrophytes: Eichhornia crassipes, Limnocharis flava, and Neptunia oleracea. The study found that all three species are rich in nutrients and minerals and can serve as beneficial supplements for human consumption. A lot of the information has been updated since last version.

Experimental design

The authors addressed all my concerns on experimental design.

Validity of the findings

the authors has corrected result part and added additional information.

Additional comments

In general, the authors have made notable progress in their use of the English language and have effectively addressed all of my concerns.

---

## Round 0.3 · accepted · Accept

The authors addressed all of the reviewers' comments, including the Editor's note that "English proofreading (mainly for grammar) should be done before publication" was accomplished. This manuscript is ready for publication